# Different Reactions in Each Enterotype Depending on the Intake of Probiotic Yogurt Powder

**DOI:** 10.3390/microorganisms9061277

**Published:** 2021-06-11

**Authors:** Songhee Lee, Heesang You, Minho Lee, Doojin Kim, Sunghee Jung, Youngsook Park, Sunghee Hyun

**Affiliations:** 1Department of Biomedical Laboratory Science, Graduate School, Eulji University, Dongil-ro 712, Uijeongbu-si 11759, Korea; song-1107@naver.com; 2Department of Senior Healthcare, Graduate School, Eulji University, Dongil-ro 712, Uijeongbu-si 11759, Korea; yhs1532@nate.com; 3Department of Food Science and Service, College of Bio-Convergence, Eulji University, Sansung daero 553, Seongnam-si 13135, Korea; minho@eulji.ac.kr; 4Department of Biomedical Laboratory Science, College of Health Sciences, Eulji University, Sansung daero 553, Seongnam-si 13135, Korea; djkingdom@hanmail.net; 5Department of Internal Medicine, College of Medicine, Eulji University, Dunsan-seo 95, Daejeon-si 35233, Korea; jsh@eulji.ac.kr; 6Department of Gastroenterology, Nowon Eulji University Hospital, Eulji University School of Medicine, Hangeul Biseok-ro 68, Seoul 01830, Korea; pys1109@eulji.ac.kr

**Keywords:** gut microbiota, 16S rRNA, probiotics, enterotype

## Abstract

Probiotics can be used as a nutritional strategy to improve gut homeostasis. We aimed to evaluate the intestinal microbiota profile of 18 subjects after ingestion of probiotic yogurt powder (PYP) based on enterotype. The subjects were classified into three enterotypes according to their microbial community: *Bacteroides* (*n* = 9, type B), *Prevotella* (*n* = 3, type P), and *Ruminococcus* (*n* = 6, type R). We performed controlled termination in a transient series that included a control period of three weeks before probiotic intake, PYP intake for three weeks, and a three-week washout period. Fecal microbiota composition was analyzed by sequencing the V3–V4 super variable region of 16S rRNA. Based on the Bristol stool shape scale, abnormal stool shape improved with PYP intake, and bowel movements were activated. The abundance of *Faecalibacterium*, *Eggerthella*, and *Leuconostoc*, which ferment and metabolize glucose, showed a strong correlation with type B *Bacteroides*, and glucose metabolism improvement was observed in all type B subjects. Alkaline phosphatase was significantly improved only in type B. In addition, the abundance of type B *Bacteroides* showed a negative correlation with that of *Lactobacillus*. The abundance of *Streptococcus*, *Agathobacter*, and *Christensenella*, which are involved in lipid metabolism, showed a strong correlation with that of type P *Prevotella*, and triglyceride metabolism improvement was observed in all type P subjects. The gut microbiota showed only short-term changes after PYP intake and showed resilience by returning to its original state when PYP intake was interrupted. In summary, the different responses to PYP intake may result from the different enterotypes and associated strains; therefore, the probiotic composition should be adjusted based on the individual enterotype.

## 1. Introduction

The intestinal microbiota contains the highest number and concentration of microorganisms living in the human body [1]. As confirmed in several studies over the past decades, the gut microbiota not only plays an important role in the digestion of food, but is also responsible for the development of several metabolic disorders, including obesity, nonalcoholic liver disease, type 2 diabetes, and heart disease. Several recent studies have explored the potential role of the gut microbiota, its association with fecal, blood, and urine metabolites, and its determinants [1,2].

The intestinal microbiota is unique to each individual owing to the diversity of strains, differences in the growth rate of each microbe, structural variations in microbial genes, and differences in the environment [3,4,5]. Despite the high degree of variability among individuals, a recent analysis of the gut microbiota classified it into three types distinguished by distinct genera, each one dominated by *Bacteroides*, *Prevotella*, or *Ruminococcus* [6]. The essence of enterotyping is stratifying the entire human gut microbiome; it serves as a dimensional reduction of the global microbiome variations into several categories. These categories, called enterotypes, were originally reported as “highly populated areas in a multidimensional space of community composition”, which means that they are not as sharply distinct as, for example, blood types [6]. In addition, clear evidence and criteria for enterotyping are lacking and appear to be independent of nationality, sex, age, or body mass index (BMI) [7]. However, these enterotypes are known to be affected by short- and long-term diets [8,9].

Statistics Korea, the Korean statistics authority, reported that in 2016, the population of individuals aged 65 or older was 13.2% of the total population, which was nearly twice that in 2000 (7.2%) [10]. Moreover, the aging rate in Korea is the highest worldwide. It is estimated that by 2030, the life expectancy of over 50% of Korean women will be 90 years, and the key to improving longevity at an old age is extending life expectancy [11]. Studies on age have revealed that some diseases in older people are associated with changes in the intestinal microbiota [12]. Therefore, studies investigating changes in the intestinal microbiota with aging are required [12,13]. Menopause in women occurs between the ages of 45 and 55 and involves cessation of menstruation and loss of ovarian reproductive function [14]. The changing levels of female sex hormones during this period affect the composition of the microbiome in different parts of the body, especially the gut. Interestingly, the presence or absence of estrogen may alter the gut microbiome and corresponding disease pathways [15,16]. Thus, an imbalance in the gut microbiota, called dysbiosis, is widely associated with metabolic and immune diseases. To date, few studies have investigated changes in the gut microbiome in the absence of estrogen, that is, metabolism in postmenopausal women. Many questions remain unanswered because it is unclear which changes in the gut microbiome cause the overall changes in postmenopausal women. Probiotics and prebiotics can be used in conjunction with postmenopausal hormone therapy and may alleviate the side effects of hormone replacement therapy [17].

A healthy gut ecosystem can be altered due to alterations in microbial composition, primarily attributed to dietary patterns (vegetarian and western), probiotics, and lifestyle [18,19,20]. Western diet is characterized by a high content of unhealthy fats, refined grains, sugar, salt, alcohol, and other harmful elements, along with a reduced consumption of fruits and vegetables [21,22]. Western diet promotes a pathological microbiota status, leading to an increase in the *Firmicutes*/*Bacteroidetes* ratio. This induces critical changes in both the gut microbiota and immune system, negatively affecting the gut integrity, and thus promoting local and systemic chronic inflammation [21,22,23,24]. Mediterranean diet is a nutritionally recommended dietary pattern characterized by a high intake of fruits, vegetables, legumes, nuts, and minimally processed cereals; moderately high consumption of fish; low intake of saturated fat, meat, and dairy products; and regular, but moderate, consumption of alcohol [25]. The consumption of fruits, vegetables, beans and plants in the Mediterranean diet is likely facilitated by certain bacteria belonging to the phyla Firmicutes and Bacteroides, which can break down carbohydrates that the host cannot digest [25]. The Korean diet involves proportionally high consumption of vegetables, high consumption of whole grain rice, moderate to high consumption of beans and plants and fish, and low consumption of red meat. The traditional Korean diet is different from the Mediterranean diet, but they both involve a high consumption of vegetables and low consumption of processed meat [26,27]. The unique feature of the Korean diet is the high consumption of fermented foods. Korean diets include at least one fermented food, such as kimchi, in a meal consisting of rice, soup, and various side dishes. The intake of *Lactobacillus* contained in kimchi has been suggested to have potential probiotic effects [26,28,29]. Recent studies have shown that high animal protein and saturated fat consumption is related to the Bacteroides enterotype, linking the meat-rich Western diet to this enterotype. In contrast, *Prevotella* is associated with high carbohydrate and monosaccharide consumption, indicating its relationship with a typical carbohydrate-based diet in an agricultural society and linking it to vegetarianism, but there are still no clear criteria for judging that it is related to vegetarianism [30,31].

Although about half of Korean adults and 78% of senior citizens (65 years and older) maintain a traditional Korean diet, the country’s recent rapid economic development has blurred its diet’s boundary with other diets, such as the Mediterranean diet [24,25,26,27,28,29,30,31,32,33,34]. In addition, most Mediterranean diets are described using subjective terms, such as moderate, high, without specific suggestions for serving size and criteria for the amount of additives in diets, such as sauce, condiments, tea, coffee, salt, sugar, and honey [35]. So far, most studies that have examined links between intestinal microbial communities based on diet have focused on Western or Mediterranean diet and have been conducted primarily in European and American populations. Conversely, little is known about the relationship between intestinal microbial groups based on Korean diet [15,36,37]. It is thus important to observe changes in intestinal microorganisms due to probiotic intake by Koreans consuming a diet with a high content of whole grains and vegetables.

Human gut microbiota analysis can be performed using novel tools such as next-generation sequencing (NGS) in culture-dependent methodologies [38]. The most widely used amplicon analysis involves 16S ribosomal RNA genes [5,39,40,41,42]. NGS is useful for sequence assembly, mapping, and analysis of the vast amount of data generated, so metagenomic or 16S rRNA gene sequencing data are required for long-form analysis. This can be the most efficient tool for analyzing complex microbial communities [43].

Finally, it is necessary to recognize that the intestinal microbial community is constantly changing with a high diversity and dynamic state from individual to individual.

The use of probiotics and prebiotics is a nutritional strategy to improve gut homeostasis [12]. Prebiotics are selectively fermented ingredients resulting in specific changes in the composition and activity of the gastrointestinal microbiota, thus providing health benefits for the host [44]. They are complex carbohydrates or compounds and are not digested and absorbed in the small intestine but play a variety of roles in the intestine. Probiotics are defined as live microorganisms that, when administered in adequate amounts, confer health benefits to the host [45].

We hypothesized that short-term consumption of probiotic yogurt powder (PYP) not only improves intestinal microbial and bowel activity, but also improves metabolism index values in the blood of healthy subjects. In addition, changes differ according to the enterotype, and each enterotype metabolizes in correlation with numerous other strains. In this study, we observed the effects of short-term consumption of PYP on metabolic indicators, bowel movements, and intestinal microbial communities in 18 middle-aged and elderly Korean women.

## 2. Materials and Methods

### 2.1. Study Schedule and Design

This study was a controlled-time series experiment and designed as a termination, panel, and pilot study. The flow of this study can be divided into three main periods: (1) a 3-week control period in which the study participants maintained their usual diet excluding probiotics, (2) a 3-week experimental period with PYP consumption, and (3) another 3-week control period in which the study participants maintained their usual eating habits excluding probiotics. The starting point for each period was called (1) baseline, (2) after PYP intake, and (3) washout (Figure 1).

### 2.2. Participant Recruitment and Screening

The study involved 18 volunteers, all of which were women. The recruitment of volunteers was conducted at the Mirae Seum Senior Complex in Seongnam, Gyeonggi-do. All participants provided written consent and were given the option to withdraw from the study at any time. The protocol was approved by the Eulji University Internal Review Board (IRB No. EUIRB 2019-53). The inclusion criterion was healthy women over 50 years old (Table 1). Exclusion criteria included alcoholism or more than 420 mL soju (Korean distilled spirits with 19% alcohol content) intake per day, chronic smoking (more than 20 cigarettes per day), cardiovascular disease, cerebrovascular disease, pancreatitis, liver disease, kidney disease, cancer, thyroid disease, dementia, Parkinson’s disease, depression, anorexia/multiple sclerosis, or multiple sclerosis. Participants were aware that they would be excluded from the study if they consumed other probiotics that affected gut health except for PYP during the total study period. In addition, participants were asked to report any newly prescribed concomitant medications, side effects, or other comments during the study period. A questionnaire was administered to all participants to collect information necessary for the study, such as disease, diet, and bowel movements, at baseline, and after time point (Appendix A).

### 2.3. YP Supplements and Intake Method

The PYP used in this study was Low Fat Yogurt Powder (FreshEtto, Seoul, Korea) and its quality was consistent with Codex Alimentarius standards. The PYP contained more than 1 million colony-forming units (CFU)/g of total lactic acid bacteria. The sub-components consisted of 10^6^ CFU/g of *Streptococcus thermophilus* and 10^4^ CFU/g of *Lactobacillus bulgaricus*.

All participants were instructed to consume milk (200 mL) with PYP (20 g) within 30 min of breakfast once a day. In order to ensure compliance of the participants, we distributed a daily self-intake table and checked it by phone every week.

### 2.4. Sample Collection

The anthropometric information included measurements of weight, height, and BMI. The weight was measured while fasting. Height was measured barefoot, and weight was measured with the coat off. Waist circumference was measured midway between the lower rib edge and the iliac ridge using a non-sagging measuring tape. The BMI was calculated as weight (kg) divided by height (m^2^) and interpreted according to Word Health Organization 2000 guidelines [46,47].

Blood pressure was measured using a sphygmomanometer after the participant rested for 5 min in a sitting position. For blood parameters, blood pressure was measured, and a fasting blood sample was taken from the anterior vein of the arm. Blood samples were mixed in Ethylene-Diamine-Tetraacetic Acid (EDTA) Vacutainer Blood Collection Tubes (Becton Dickinson, Franklin Lakes, NJ, USA). Plasma and red blood cells were separated from the samples by centrifugation (1500× *g*, 4 °C, 15 min). For the determination of serum biochemical parameters, the blood samples were sent to Seongnam City Central Hospital for immediate analysis using standard laboratory methods and certified assays. Automated analyzers (Roche Diagnostics, Seoul, Korea) were employed for measuring fasting blood glucose, triglycerides (TG), cholesterol, high- and low-density lipoprotein, lactate dehydrogenase, creatinine, alkaline phosphatase (ALP), and C-reactive protein.

For urinalysis, dipstick urine analysis was performed on each urine sample using Urine Reagent Strip, ComboStic 10 (DFIcare, Gyeongsangnam-do, Korea). The laboratory technician performed the test within 2 h of sample collection, according to the manufacturer’s instructions. The results were read within 5 min [48].

We provided stool containers to the participants before each visit and the stool samples were freshly collected (0.25 g) the night or morning before the visit. They were stored for less than 4 h in a 4 °C household refrigerator before being transferred to the laboratory. DNA was then extracted from the samples. The samples were then stored at −80 °C until further analysis.

### 2.5. Gut Microbiota Analysis

#### 2.5.1. Fecal DNA Extraction

DNA extraction was performed using QIAamp PowerFecal Pro DNA Kit (Qiagen, Hilden, Germany) following the manufacturer’s instructions. In brief, a 250-mg aliquot of the fecal sample was transferred to a dry Bead Tube provided in the kit. Next, 800 µL of C1 solution was added, and the sample was vortexed at maximum speed for 10 min. The rest of the protocol was performed following the manufacturer’s instructions. DNA was eluted in 65 µL of C6 elution buffer solution. The extracted DNA samples were stored at −80 °C until library preparation and sequencing [49].

#### 2.5.2. Polymerase Chain Reaction (PCR) Amplification of the 16S rRNA Genes

The extracted DNA was used as a template for PCR amplification of the V3–V4 region of bacterial 16S rRNA genes using the following adapter sequences, index sequence, and general-purpose primers: 341F (5′-CCT ACG GGN GGC WGC AG-3′) with a sample-specific 6–8 bp tag sequence and 805R (5′-GAC TAC HVG GGT ATC TAA TCC-3′). PCR was performed using the Platinum PCR SuperMix High Fidelity system (Thermo Fisher Scientific, Waltham, MA, USA) using 2.5 ng of template DNA and each primer at a final concentration of 50 nM in a 27 µL final reaction volume. PCR was performed under the following cycling conditions: 94 °C for 3 min, followed by 30 cycles of 94 °C for 30 s, 50 °C for 30 s, and 72 °C for 30 s. The amplicon libraries were further purified to remove residual primer dimers and any contaminants using the Agencourt AMPure XP DNA Purification Kit (Beckman Coulter, Brea, CA, USA), following the manufacturer’s instructions. Samples were eluted in 15 µL low-EDTA Tris EDTA buffer. The DNA concentration, quality, and amplicon library concentrations were assessed using the dsDNA HS (High Sensitivity) Assay Kit on the Qubit 4 Fluorometer instrument (Thermo Fisher Scientific, Waltham, MA, USA). The fragment size and quality of the pooled DNA were assessed using Agilent 2100 Bioanalyzer system (Agilent Technologies, Palo Alto, CA, USA). The enriched particles were loaded onto the Ion 530 Chip Kit (Thermo Fisher Scientific, Waltham, MA, USA), and sequencing was performed on Ion GeneStudio S5 (Thermo Fisher Scientific, Waltham, MA, USA) according to the manufacturer’s instructions [50,51,52,53]. After PCR amplification, paired-end sequencing was performed using Ion GeneStudio S5 next-generation sequencing system (Thermo Fisher Scientific, Waltham, MA, USA).

#### 2.5.3. 16S rRNA Gene Sequencing Data Processing and Identification of Microbial Taxa

The FASTQ file, the raw data of 16S rRNA sequences, was obtained using the Torrent Suite Software version 5.14.1.1. (Thermo Fisher Scientific, Waltham, MA, USA). The 16S rRNA workflow module in the EzBioCloud software (ChunLab, Seoul, Korea) was used to classify individual reads by combining the Basic Local Alignment Search Tool with the curated Greengenes Database, which contains a high-quality library of full-length 16S rRNA sequences. The reads were excluded from the analysis if they were shorter than 500 bp or inappropriately paired. Chimeras were removed from the sequence data. Sequences were clustered into operational taxonomic units at 97% identity using QIIME pick_open_reference_otus.py, the Greengenes 13.5 reference database, and UCLUST algorithm; 3,612,748 total read counts and 66,902 average counts per sample were obtained.

### 2.6. Statistical Analyses

Statistical analyses were performed using GraphPad Prism version 8.3.1 for Windows (GraphPad software, San Diego, CA, USA) and SPSS version 20.0 (SPSS, Chicago, IL, USA). Utilizing the hierarchical structure of taxonomic classification, tree analysis was performed with quantitative (using medium abundance) and statistical (using non-parametric Wilcoxon Rank Sum test) hits to depict taxonomic differences between microbial communities or abundance profiles of samples, groups, or the whole microbial community. The changes (mean ± standard deviation) in individual biochemical indices and microbiota between the baseline, after-PYP-intake, and washout periods were assessed with a paired *t*-test or a Wilcoxon signed-rank test according to the data distribution. Differences were considered statistically significant at *p* < 0.05. Spearman’s correlation coefficient (rho, ρ) was used to determine the association between intra-individual similarities of the microbiota after PYP intake. Principal component analysis (PCA), a linear dimension reduction method, was used to determine major changes at the genus level, and data were compressed into several informational features that allowed for two-dimensional visualization of sample similarity. Beta diversities were analyzed using the EzBioCloud dashboard (ChunLab, Seoul, Korea) [54]. PCA was conducted using the Jensen–Shannon divergence method [55]. The Ion Reporter suite (Thermo Fisher Scientific) was used to filter polymorphic variants. Statistical taxonomic comparisons were performed using linear discriminant analysis effect size (LEfSe) analysis on the Galaxy Hutlab online platform (Hutlab, Boston, MA, USA; https://huttenhower.sph.harvard.edu/galaxy, accessed on 26 July 2016) [56,57]. LEfSe is a tool developed to identify biomarkers between two or more groups using relative abundances, and linear discriminant analysis is used to model the difference between classes of data. Correlations between OTUs taking this specific problem into account were evaluated using SparCC, a tool that assumes a sparse network and iteratively performs a log ratio transformation to identify outlier taxa for correlation. Correlation and pattern analysis were performed using SparCC.

## 3. Results

### 3.1. Basic Demographic Characteristics of Participants

All participants were menopausal women with an average age of 66 years and were non-smokers. The study protocol and design are summarized in Figure 1. Most of the participants engaged in light physical exercise such as walking for about 30 min every day and usually consumed a mixed Korean or Mediterranean diet. 

### 3.2. Blood Parameters and Human Body Measurements

We observed changes in blood parameters in all participants (Table 2). We found that glucose levels decreased by 3.5% after PYP intake, and at the time of washout, they returned to normal. Glucose levels before PYP intake and at washout point did not differ significantly. No improvement was observed for most of the chemical parameters (Appendix A). As a safety indicator, hematological parameters showed similar values at all time points, and no parameters changed outside the normal range (Appendix A).

### 3.3. Changes in Stool Conditions

We observed changes in stool type, bowel activity, and beneficial bacteria at each time point (Figure 2). In the Bristol stool form scale standard, type 4 stool samples are ideal. The stool type of the participants significantly improved from 4.8 at baseline to 4.2 after PYP intake, and then increased to 4.4 at washout. The defecation activity observed for one week increased from 5.7 to 5.9. Analysis of the effects of PYP intake on the abundance of several beneficial bacteria revealed that among the beneficial bacteria, the abundance of *Bifidobacterium adolescentis*, *B. longum*, *Lactobacillus rogosae*, and *L. sakei* increased.

### 3.4. Phylogenetic Tree Analysis According to PYP Intake

We investigated the overall systemic structure of microorganisms according to the time of PYP ingestion (Figure 3). Utilizing the hierarchical structure of taxonomic classification, we performed a heat tree analysis to statistically depict the differences between microbial communities or the abundance profile of samples. We compared the baseline and after-PYP-intake timepoints. Compared to the after-PYP-intake timepoint, an abundance of *Blautia*, *EU779302*_*s*, *DQ794680*_*s*, *AF349417*_*s*, *DQ799511*_*s*, *PAC001200*_*g*_*uc*, *Anaerostipes*_*uc*, *DQ798794*_*s*, *Ruminococcus*_*g4*_*uc*, *Eubacterium*_*g5*_*uc*, *Dorea*_*uc*, *PAC001138*_*s*, *PAC001138*_*s*, and *PAC001138* was observed at baseline. Next, after-PYP-intake and washout points were compared. After PYP intake, the abundance of bacterial strains that were abundant at baseline decreased and the abundance of *PAC001037*_*s*, *PAC001316*_*s*, and *PAC002119*_*s* increased. When baseline and washout timepoints were compared, only *JH815484*_*s* was observed in the heat tree. In other words, heat tree analysis showed that the intestinal microbial system partially changed after PYP intake. In addition, the changed intestinal microbial system was restored to its original state when ingestion was stopped (at the washout timepoint).

### 3.5. Enterotyping

We performed enterotyping of all subjects (Figure 4). Principal coordinate analysis was performed based on the Bray–Curtis index. The bacteria were classified into three genera: *Bacteroides* (type B), *Prevotella* (type P), and *Ruminococcus* (type R). Therefore, the microbiota was classified into three enterotypes in the same manner as other previous studies (Figure 4A,B). Nine subjects had type B microbiota, three had type P, and six had type R. In order to increase the accuracy of enterotyping, we once again observed the abundance of the main strains of each enterotype in detail. Based on analysis of variance, we analyzed classical univariate statistical comparisons and plot of features with statistically significant differences between enterotypes (Figure 3C).

### 3.6. Changes in Specific Blood Index Values Based on Enterotype

Changes in blood index values were observed for each enterotype (Figure 5). Blood glucose levels decreased from 84.83 to 81.83 mg/dL (*p* = 0.027) after PYP intake and returned to 85.18 mg/dL (*p* = 0.012) at washout, when PYP intake was stopped, in all subjects. After enterotyping, blood glucose levels significantly decreased after PYP intake only in type B subjects (*p* = 0.0027). Moreover, on average, ALP levels decreased from 74.67 IU/L to 71.78 IU/L (*p* = 0.034) in all subjects after PYP ingestion and returned to 74.26 IU/L (*p* = 0.242) upon washout after PYP intake was stopped. However, according to enterotyping, ALP levels were significantly reduced only in type B subjects after PYP intake (*p* = 0.0105). Further, on average, AST levels increased from 25.0 IU/L to 25.22 IU/L (*p* = 0.791) in all subjects after PYP intake. In contrast, after enterotyping, only the AST levels of type P subjects did not show an increase, but exhibited maintained or decreased levels. In addition, on average, TG levels increased from 118.06 mg/dL to 120.83 mg/dL (*p* = 0.783) after PYP ingestion, but decreased at washout when PYP intake was stopped (135.06; *p* = 0.168). Enterotyping revealed that the TG levels of type P subjects significantly improved after PYP intake (*p* = 0.0238).

### 3.7. Enterotype-Based Changes in Intestinal Microflora According to PYP Intake

We performed heatmap analysis to observe changes in the abundance of microorganisms according to each enterotype (Figure 6). We observed changes between the three enterotypes in the bacterial abundance at the family level at the three time points. *Bacteroidaceae* and *Christensenellaceae* were relatively abundant in type B subjects; *Prevotellaceae* and *Erysipelotrichaceae* were abundant in type P participants; and *Aerococcaceae*, *Ruminococcaceae*, and *Bifidobacteriaceae* were abundant in type R enterotype. However, as the subgroups diversified, individual diversity emerged, so we focused on microbes at the genus level and performed heatmap analysis again. The relative abundance of microorganisms was compared between the three enterotypes. In type B, *Subdoligranulum*, *Eggerthella*, *Syntrophococcus*, *Anaerotignum*, and *Bacteroides* were relatively abundant; in type P, *PAC001398*_*g*, *Prevotella*, *Holdemanella*, and *Howardella* were abundant; in type R, *PAC000195*_*g*, *PAC001138*_*g*, and *Lachnospira* were abundant.

### 3.8. Analysis of Significant Strains Showing Correlation by Enterotype

We divided each enterotype based on three strains, and found that each enterotype strain was related to numerous other strains (Figure 7).

We observed 24 genera that were significantly related to type B *Bacteroides*, 12 with a positive correlation and 12 with a negative correlation (Figure 7A,B,G). *Faecalibacterium*, *Eggerthella*, *Granulicatella*, *Sellimonas*, *Faecalicatena*, *Leuconostoc*, *Streptococcus*, *Eubacterium*_*g4*, *Clostridium*_*g24*, *Longicatena*, *Veillonella*, and *Rothia* showed a significant positive correlation with *Bacteroides*, while *PAC001457*_*g*, *Syntrophococcus*, *Sporobacter*, *PAC001296*_*g*, *Eubacterium*_*g23*, *PAC0001600*_*g*, *PAC000661*_*g*, *PAC001041*_*g*, *PAC001207*_*g*, *KE159605*_*g*, *Lactobacillus*, and *PAC001270*_*g* showed a significant negative correlation with *Bacteroides*.

Similarly, we observed 25 genera that were significantly related to type P *Prevotella*, 14 with a positive correlation and 11 with a negative correlation (Figure 7C,D,H). *Streptococcus*, *Dialister*, *PAC001138_g*, *PAC001276_g*, *Senegalimassilia*, *Megasphaera*, *Sutterella*, *Lachnospira*, *PAC000197*_*g*, *Collinsella*, *Mitsuokella*, *PAC001398*_*g*, *Megamonas*, and *Agathobacter* showed a significant positive correlation with *Prevotella*, whereas *PAC001212*_*g*, *PAC001437*_*g*, *PAC001457*_*g*, *Christensenella*, *Eubacterium*_*g4*, *PAC001236*_*g*, *Paludicola*, *PAC001283*_*g*, *PAC001637*_*g*, *NHOC_g*, and *Desulfovibrio* showed a significant negative correlation with *Prevotella*.

We also observed 25 genera that were significantly related to type R *Ruminococcus*, 13 with a positive correlation and 12 with a negative correlation (Figure 7E,F,I). There was a significant positive correlation with *PAC000196*_*g*, *CCMM*_*g*, *Faecalibacterium*, *Eubacterium*_*g17*, *Eubacterium*_*g23*, *PAC000661*_*g*, *Turicibacter*, *LLKB*_*g*, *Parasutterella*, *Eubacterium*_*g24*, *PAC000740*_*g*, *PAC001207*_*g*, and *Romboutsia*. Meanwhile, *Clostridium*_*g24*, *Paraprevotella*, *Leuconostoc*, *Raoultella*, *Alistipes*, *Pantoea*, *Parabacteroides*, *Anaerofustis*, *Klebsiella*, *Pseudoflavonifractor*, *Enterobacteriaceae*, and *Ruminococcus*_*g5* were significantly negatively correlated with *Ruminococcus*.

## 4. Discussion

In this study, we found that (1) PYP intake ameliorated abnormal stool morphology in BSFS and promoted bowel activity; (2) PYP intake was associated with an improvement in a small number of blood indicators; (3) PYP intake altered the intestinal microbial system; (4) the intestinal microbiota of all participants was classified into three enterotypes based on the similarity of intestinal composition; here, type B was identified as *Bacteroides*, type P as *Prevotella*, and type R as *Ruminococcus*; (5) type B significantly improved blood glucose and ALP levels, and type P was effective in significantly improving TG levels; (6) individual intestinal microbes vary, but strains with varying abundance differed in each enterotype according to PYP intake; and (7) the main strains of each enterotype had a close correlation with a wide variety of bacteria.

Although most Koreans have recently become Westernized, they still maintain high vegetable consumption and consumption of soybeans and fish. The Mediterranean diet and the Korean diet are similar in that they both involve consuming high amounts of vegetables and beans and fish. Based on this, we divided the subjects into a Mediterranean diet, consuming small amounts of meat and processed meat, and a mixed Korean diet, consuming kimchi at every meal and a small amount of meat. The main purpose of this study was to observe how PYP intake exhibits different effects when enterotyping a Korean elderly woman who has a specific diet (mainly a Korean diet or a Mediterranean diet similar to a Korean diet). 

This study recommended maintaining one’s lifestyle for the total duration of the study. However, the basic characteristics investigated at baseline may have had some effect on PYP intake. Smoking has an adverse effect on the clinical process of Crohn’s disease and can prime neutrophils; however, in this study, all the subjects did not have a history of smoking, and thus no abnormal values were obtained in the assessment of safety parameters [58,59,60]. Acetaldehyde, a product of alcohol degradation, increases intestinal permeability by itself, indirectly causing microbial dysbiosis, and increases Lipopolysaccharides (LPS) levels [61,62]. Most of the subjects in this study did not normally consume alcohol, and the four respondents who said they consume alcohol also took it once a week. Most of the subjects exercised lightly for more than 30 min daily, as moderate and intense exercise has a better effect on gastrointestinal health than non-exercise [63]. A study evaluating the effect of women’s intake of probiotic yogurt on oxidative stress and inflammatory factors found that women who consumed probiotic yogurt showed an increase in antioxidant levels [64].

Before classifying all subjects into the various enterotypes, we observed changes in fecal type, defecation, and specific beneficial bacteria in Figure 2. After enterotyping, we observed these parameters again according to enterotype (Appendix A). Fecal type was observed to be significantly improved, especially in the R type (*p* = 0.034, Appendix A). The changes in the four specific strains show in Figure 2 were observed by enterotype through reads (Appendix A). There is a difference in the degree of change by strain for each enterotype, but all four strains increased after PYP intake (after timepoint) in all enterotypes. In type B, the abundance of *Bifidobacterium adolescentis* and *Bifidobacterium longum* was significantly increased. In type P, in particular, the abundance of *B. longum* and *Lactobacillus rogosae* was significantly increased. In type R, the abundance of *B. adolescentis* and *B. longum* was significantly increased.

The fact that the strains associated with each of the three enterotypes differed according to PYP intake may be because each enterotype responds differently to PYP intake. Some of the numerous strains involved are worth noting. *Bacteroides* is a genus of great interest for intestinal microbiota investigation owing to its strong adaptability properties in the host and its fundamental advantages. It has been shown to be much more stable than *Firmicutes* [65]. The main strength of *Bacteroides* is breaking down powerful polysaccharides that provide both nutrients and energy to the host, producing short-chain fatty acids (SCFAs) from the daily ingested food or deriving them from intestinal mucus by ingestion of indigestible vegetable or animal glycans [66]. Recently, the European Commission approved the study of *Bacteroides* as a starter in the fermentation of pasteurized milk products according to EU Novel Foods Regulation No. 258/97 [67]. In our study, type B *Bacteroides* showed positive and negative correlation with 12 and 12 genera, respectively.

*Faecalibacterium* in type B subjects showed the greatest positive correlation with *Bacteroides*, and *Faecalibacterium prausnitzii* was the only species observed. In previous studies, *Faecalibacterium* had been shown to have a strong positive correlation with glucose oxidase, thus improving enzymatic activity and nutrient digestibility in the small intestine [68]. *F. prausnitzii* is a major component of the gut microbiota and is the most important butyrate-producing bacteria in the human colon. When its levels decrease, inflammation increases; it is also negatively associated with inflammatory bowel disease and type 2 diabetes. The recently isolated *Faecalicatena* and *Eggerthella* have also been associated with glucose fermentation [69,70]. In addition, *Leuconostoc* regulates blood sugar and insulin in diabetic mice [71]. In particular, in type R enterotype, *Leuconostoc* had a negative correlation with *Ruminococcus*. It is expected that *Leuconostoc* abundance is the most pronounced in type B enterotype. As the results of this study showed that *Leuconostoc* abundance significantly decreased only in the type B enterotype, the causal relationship according to the abundance of these strains can be predicted. It should be noted that these strains were clearly observed in all subjects, but they had a strong positive correlation with *Bacteroides* in type B enterotype. Therefore, probiotics using *F. prausnitzii* as a key ingredient show potential as a therapeutic agent for improving intestinal dysbiosis in subjects with type B enterotype [72].

In this study, ALP levels significantly improved only in type B enterotype. ALP is present in hepatocytes in the gallbladder duct and is mainly elevated when bile excretion is impaired. According to a previous study, high abundance of *Veillonella* with high ALP levels is observed in patients with primary biliary cholangitis (PBC) [73,74]. In addition, stool treatment for PBC-inflammatory bowel disease (IBD) leads to a decrease in ALP levels [75]. The pattern of treated PBC-IBD patients and the results observed in this study are very similar; thus, the possibility of a causal relationship with PYP intake cannot be ruled out. In addition, *Bacteroides* had a negative correlation with *Lactobacillus*. *Lactobacillus* is a major strain that has conventionally been used in probiotics. However, in this study, there was a negative correlation with type B *Bacteroides*, suggesting that the intake of probiotics consisting only of *Lactobacillus* for subjects with type B enterotype may be ineffective.

It has been suggested that *Prevotella* is promoted by dietary plant polysaccharides but not animal polysaccharides. This shows the correlation between a vegetable-rich diet and the abundance of *Prevotella*. Therefore, to maintain an abundance of *Prevotella*, dietary plant polysaccharides are continuously required, and the absence of plant polysaccharides can eliminate *Prevotella* from the intestinal microbial community [76]. In addition, the abundance of *Prevotella* is determined by its ability to digest complex carbohydrates and its enzymatic potential to break down cellulose and xylan in food [77,78]. Recent studies have shown that the lower the abundance of *Prevotella*, the higher the incidence of IBD [63]. In this study, the diet of all subjects with type P enterotype was vegetarian-oriented. An adequate abundance of *Prevotella* will result in an efficient utilization of diet carbohydrates and production of more SCFAs [79]. Type P *Prevotella* showed a positive correlation with 14 genera and a negative correlation with 11 genera. *Streptococcus* had the greatest positive correlation, and *S. salivarius* was found to be predominant. Previous studies have shown the involvement of *S. salivarius* in the regulation of lipid metabolism [80]. In this study, S. *salivarius* abundance was considerable in type P subjects. In addition, TG levels were found to be improved only in type P subjects when compared to other groups. Although *S. thermophilus* was not observed in this study, *Streptococcus* was the most positively correlated bacteria in type P enterotype. In addition, *S. thermophilus* is the most representative subspecies of *Streptococcus* and is difficult to observe in the stomach unless ingested as a probiotic [81,82,83,84]. It also acts synergistically with *L. bulgaricus* and has been shown to be the safest probiotic [85]. Therefore, based on the results of previous studies and this study, we suggest the therapeutic potential of probiotic formulations based on *S. thermophilus* and *L. bulgaricus*, focusing on *S. salivarius*, for subjects with type P enterotype [86]. Studies have shown that *Prevotella* and *Agathobacter* are associated with high protein intake [87,88]. *Christensenella* has shown potential as a probiotic for obesity and related metabolic disorders in other studies [45,89]. In addition, this strain forms a co-occurring network with other bacteria, and an abundance of this bacterium leads to low BMI [89]. However, in this study, type P showed a negative correlation with the abundance of *Christensenella*. Therefore, the selection of *Christensenella*, a candidate for the next generation of probiotics for subjects with type P enterotype, may need to be carefully considered.

Type R *Ruminococcus* showed a positive correlation with 13 genera and a negative correlation with 12 genera. *Ruminococcus* secretes mucolytic alpha and glucosidase, which can sensitize mucosa and mucoproteins to degradation and consumption by other co-cultured microbial species [90,91], as carbohydrate fermentation is required for their growth [92]. Recent studies have observed high levels of *Ruminococcus* in IBD patients [93]. In this study, all subjects with type R enterotype had a Bristol stool form scale level of 5 or higher. In addition, *Parasutterella*, which had a positive correlation with *Ruminococcus*, is associated with the development and progression of IBS [94]. Numerous studies on enterotypes recently revealed that the *Ruminococcus* group is an ambiguous enterotype compared to the enterotypes B and P (*Bacteroides* and *Prevotella* groups) [95,96,97,98,99]. Though enterotypes are strong among the populations according to the concept of enterotyping, which is a dimensional reduction process that stratifies the entire human intestinal microbiota to reduce the total variation of the microbial cluster into several categories [6], the question of whether the *Ruminococcus* enterotype should be discarded remains controversial [100]. In addition, in a recent study, it was found that the *Bacteroides* and *Ruminococcus* types overlap in Koreans without being separated into two independent clusters [96,101]. Some studies, including this one, only analyzed *Ruminococcaceae* or a few *Ruminococcus* genera. These inconsistencies for this enterotype suggests that it may not be well-defined by a single genus or should be classified into smaller subgroups [102].

Over the past 20 years, it has been shown that the gut microbiota contributes to the metabolic health of human hosts, and research on how it affects the host is currently ongoing. There is no clear classification standard for gut microbiota because of the enormous differences between populations, races, genders, and ages. However, classifications that can be applied to the majority of the population, such as enterotypes, are becoming more advanced. There are few studies on the association between intestinal microflora and metabolism. Individual clustering of intestinal microbial communities is a strong claim but strong evidence for it is lacking. Currently, we can predict the living environment based on the enterotype, yet it is difficult to distinguish each individual enterotype based on the living environment [95]. The enterotype comprises a concentrated area of the intestinal microflora and is not clearly delineated like blood types. Thus, the boundaries may be blurred, but differences between groups still exist. There is great diversity among individuals, and it is important to reduce this complexity and link the diversity to clinical characteristics. Enterotyping results should be considered with caution, as inconsistencies or errors in enterotyping can eventually affect the risk of disease or response to other drugs in subjects.

Our study has some limitations: (1) Since the results of this study only comprised data on subjects of the same race and region, our findings should be extrapolated to other populations with caution. However, conducting research on a homogeneous group has the advantage of significant control over the study design and raises the degree of adherence and participation. Although the number of subjects is small when we subdivided into each type, with minimal variables, these results can serve as a strong basis for more extensive studies [13]. (2) The correlation between various strains, clinical parameters, and enterotypes we demonstrated can be used for personalized treatment, but this association is not yet clear, and current data cannot explain the exact cause and effect of this relationship. It had been previously demonstrated that longitudinal studies are useful for analyzing complex gut microbiota. In addition, there is a need to further explore the variability within the intestinal microbial community at the functional level to better understand its relationship with the host [12]. (3) This study was limited to healthy subjects with no underlying health conditions and no adverse events were reported during the experimental period. Thus, all blood parameter values found to be improved in this study fluctuated within a healthy range [38]. We did not select specific bacteria to amplify and observe, but instead observed the entire microbial population using NGS [103]. The use of NGS enables the evaluation of complex microbial communities, genes, and genomes and was the most optimized method in this study [38].

## 5. Conclusions

We observed that even short-term consumption of PYP caused beneficial changes in bowel activity and intestinal microbiota. In order to observe the same results as in this study, continuous intake of PYP is a prerequisite. Each subject classified into one of three enterotypes showed significant improvement in their blood indexes. In addition, strains observed in each enterotype differed significantly, and this is expected to be the reason for different results for each enterotype. Therefore, since the intake of probiotics does not have the same effect in all subjects, probiotics must be selected by considering their type (whether types B, R, or P) in order to increase the effect.

More research focusing on the changing microbial population and its dynamic properties is required, which will further aid in the understanding of the overall community ecology of microorganisms [71].

## Figures and Tables

**Figure 1 microorganisms-09-01277-f001:**
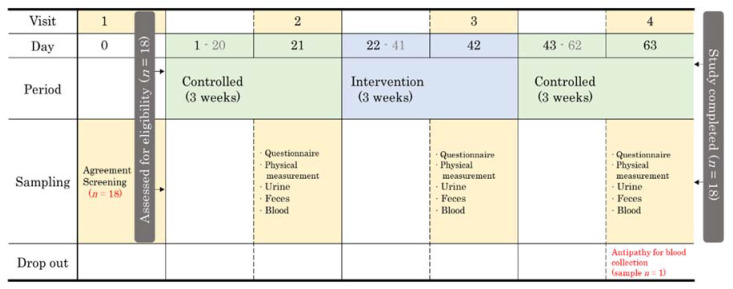
Schematic of the study design and flow.

**Figure 2 microorganisms-09-01277-f002:**
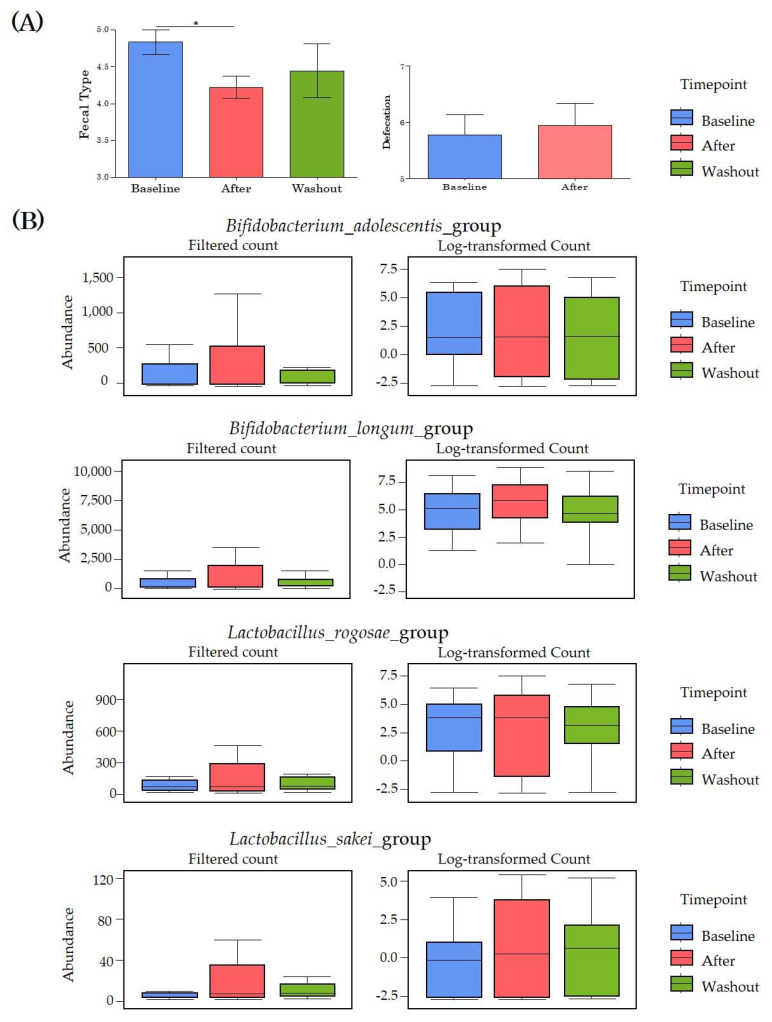
Changes in beneficial bacteria abundance and bowel activity of study participants at each time point. We observed which specific taxa were responsible for important global differences in microbial population composition between sample groups and tested all taxa individually for association with response variables. (**A**) Changes in stool type and bowel activity of participants at each time point. (**B**) Changes in the abundance of *Bifidobacterium* and *Lactobacillus* to observe changes in beneficial bacteria at each time point. * *p* < 0.05.

**Figure 3 microorganisms-09-01277-f003:**
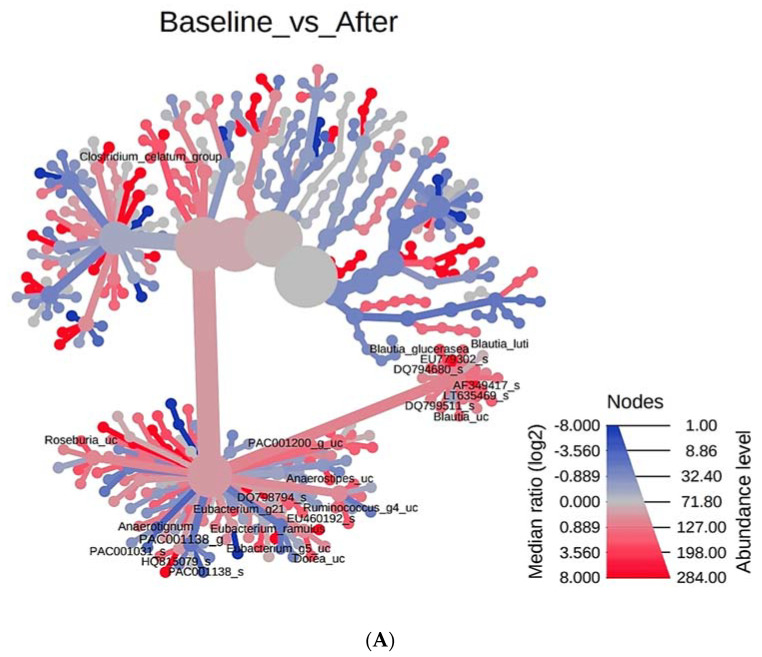
Wilcoxon single rank sum test_Heat tree. For pairwise comparisons with two or more groups, we used a heat tree, a graphing technique called a heat tree matrix. Heat tree of factor Space. Heat Trees report the effect of the presence of *vitis vinifera* roots on hierarchical structure of taxonomic classifications (median abundance, non-parameter Wilcoxon Rank Sum test). (**A**) Comparison between Baseline and After. (**B**) Comparison between After and Washout. (**C**) Comparison between baseline and washout. Each picture is marked with a different color for the level of abundance. Marked in red, it means that the value of the Median ratio (log2) increases and the abundance is high. On the other hand, indicated in blue indicates that the Median ratio (log2) is decreased and the abundance is low.

**Figure 4 microorganisms-09-01277-f004:**
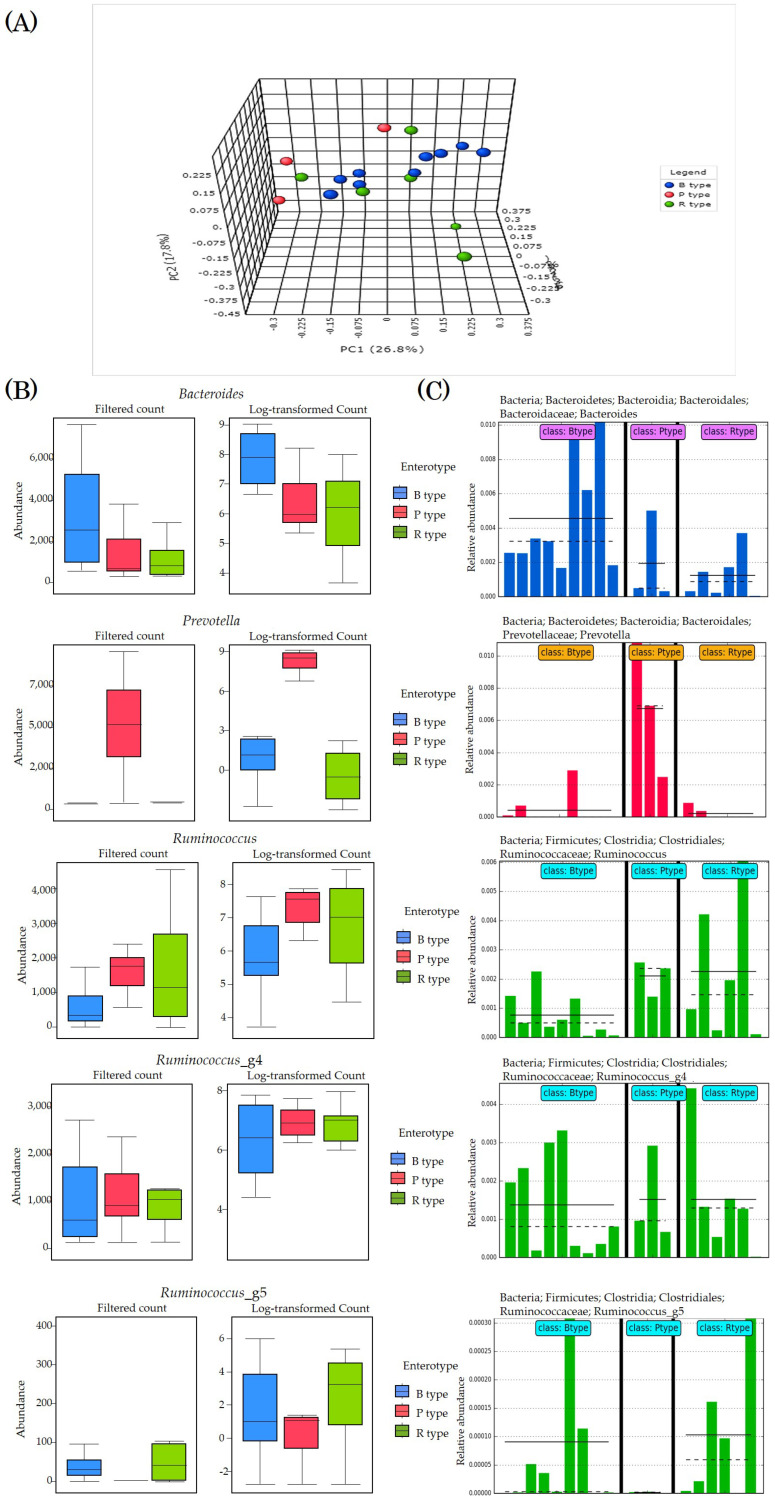
Enterotyping of all subjects. (**A**) PCoA for Genus level based on Bray–Curtis index through PERMANOVA analysis. Type B is marked with red plots (*n* = 9), P type with blue plots (*n* = 3), and R type with green plots (*n* = 6). (**B**) Classical Univariate Statistical Comparisons. Taxonomy level was Genus level, Experimental factor was Enterotype, Statistical method was Kruskal–Wallis, and adjusted *p*-value cutoff was performed at 0.05. (**C**) Plot One Feature showing the distribution of a single variable. Plot One Feature can draw histograms and histograms of specific species of biomarkers from different sample groups. The different groups in the figure are separated by a solid black line. The solid line in each group of histograms represents the average value of the sample expression level of the group, and the dashed line represents the median value of the sample expression level of the group. *Bacteroides* of B type are blue bars, *Prevetella* of P type are red bars, and *Ruminococcus* of R type (and *Ruminococcus_g4*, *Ruminococcus_g5*) are green bars.

**Figure 5 microorganisms-09-01277-f005:**
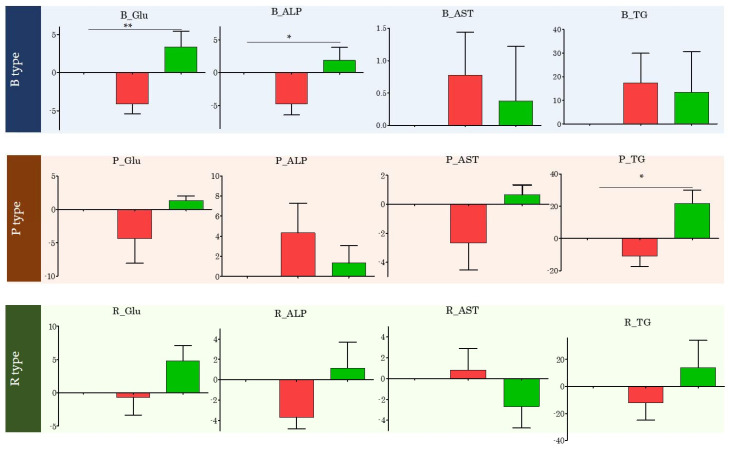
Changes in specific blood index values by Enterotype of all subjects. This picture was created to observe the change effectively. The red bar shows the change in after by setting the baseline value to 0 (reference point). The green bar shows the change in Washout by setting the value of after to 0 (reference point). One-way-ANOVA analysis was performed. * *p* < 0.05, ** *p* < 0.01.

**Figure 6 microorganisms-09-01277-f006:**
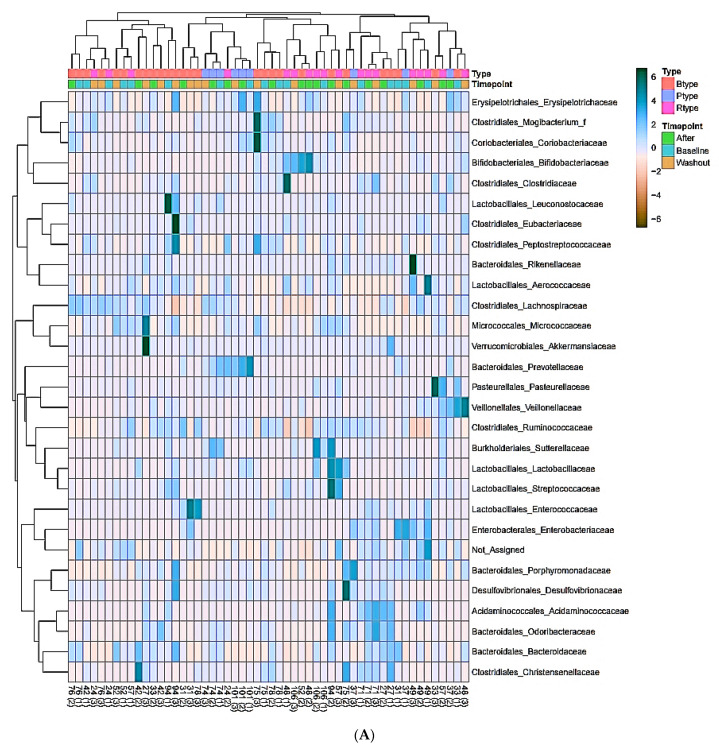
Heatmap clustering according to the Enterotype of all subjects. (**A**) The heatmap clustering shown by the enterotype for each timepoint of the subjects Taxonomy level is Family, and Order is stated in front of it. Distance measure was performed by Pearson, clustering algorithm was performed by Average, and cluster samples were performed according to the experimental factor Enterotype. (**B**) Heatmap clustering divided by enterotype at each after timepoint of subjects Taxonomy level is Genus, and Family is specified in front of it. Distance measure was performed by Pearson, clustering algorithm was performed by Average, and cluster samples were performed according to the experimental factor Enterotype. Distance measure is a method of applying a parameter in the clustering input and measuring the distance between data points, and Pearson analysis is characterized by clustering together features or samples with similar behavior. Clustering algorithm means to measure the distance between clusters, and average connection is a method of calculating the average of the distances between all points in the cluster when measuring the distance between two clusters.

**Figure 7 microorganisms-09-01277-f007:**
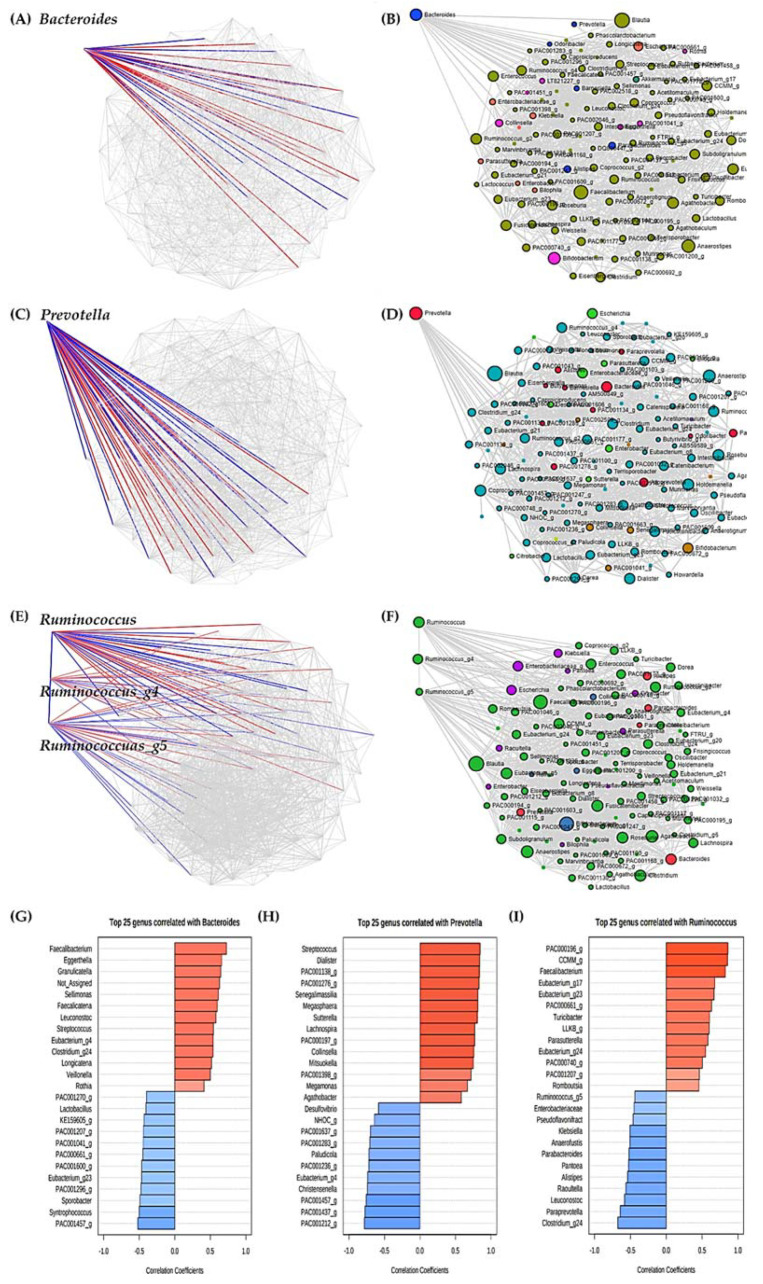
Correlation analysis between main strains of each enterotype and other strains. (**A**) SparCC between *Bacteroides* and other strains. (**B**) Line between bacteria with significant correlation with *Bacteroides*. (**C**) SparCC between *Prevotella* and other strains. (**D**) Line between bacteria with significant correlation with *Prevotella*. (**E**) SparCC between *Ruminococcus* (and *Ruminococcus_g4*, *Ruminococcus_g5*) and other strains. (**F**) Line between bacteria with significant correlation with *Ruminococcus* (and *Ruminococcus_g4*, *Ruminococcus_g5*). Algorithm performed SparCC, which assumed sparse networks and performed iterations using log-ratio transformation and identified taxa with high correlation. Permutation (SparCC) performed 100. Taxonomy level was Genus level, and *p*-value threshold was performed as 0.05. The experimental factor was Enterotype, the correlation threshold was set to 0.3, and the coloring options were set differently for each Phylum. (**G**) Overall pattern search between *Bacteroides* and other strains according to PYP intake. (**H**) Overall pattern search between *Prevotella* and other strains according to PYP intake. (**I**) Overall pattern search between *Ruminococcus* and other strains according to PYP intake. Taxonomy level was performed with Genus. Distance measure was performed by SparCC, and experimental factor was performed by timepoint.

**Table 1 microorganisms-09-01277-t001:** The characteristics of participants in the baseline.

Variables			(*n*=)
Age (Years)	50–59		2
	60–69		11
	70–79		4
	80–89		1
Menopausal transition	Menstruating		0
	Post-Menopausal		18
Smoking	No (Non-smoking)		18
	Yes (Smoking)	Current smoker	0
		Ex-smoker	0
Alcohol	No		14
	Yes	Daily	4
		Weekly	0
		Occasionally	0
Exercise	No		1
	Yes	Daily	17
		Weekly	0
		Occasionally	0
Usual diet	Western diet		0
	Mediterranean diet		7
	Mixed Korean diet		11

**Table 2 microorganisms-09-01277-t002:** Changes in chemical blood parameters at each time point of study participants.

Indices	1st (Before)		2nd (After)		3rd (Washout)		1st vs. 2nd	2nd vs. 3rd	1st vs. 3rd
	Mean	SD	Mean	SD	Mean	SD	*p*-Value		
Glucose	84.83	8.85	81.83	10.34	85.18	9.91	0.027	0.012	0.765
BUN (mg/dL)	17.47	5.46	18.09	4.15	16.02	3.73	0.358	0.010	0.152
Creatinine (mg/dL)	0.61	0.09	0.56	0.08	0.57	0.09	0.003	0.805	0.001
Uric acid (mg/dL)	4.74	0.96	4.74	0.72	4.50	0.72	1.000	0.058	0.084
TP (g/dL)	7.46	0.28	7.28	0.38	7.59	0.40	0.014	0.000	0.106
ALB (g/dL)	4.16	0.22	4.25	0.22	4.26	0.29	0.045	0.921	0.039
ALP (U/L)	74.67	17.04	71.78	16.35	74.26	16.17	0.034	0.242	0.556
AST (U/L)	25.00	7.43	25.22	8.48	24.71	6.31	0.791	0.470	0.641
ALT (U/L)	17.11	4.93	18.28	5.03	17.94	5.14	0.106	0.729	0.388
T-BILC (mg/dL)	0.63	0.21	0.62	0.11	0.61	0.14	0.683	0.833	0.721
Cholesterol (mg/dL)	214.89	38.61	218.17	44.83	225.59	42.98	0.620	0.262	0.145
Triglyceride (mg/dL)	118.06	33.68	120.83	40.30	135.06	62.40	0.738	0.168	0.080
GGT (U/L)	16.72	3.14	17.89	3.26	18.35	5.01	0.016	0.537	0.081
HDL cholesterol (mg/dL)	54.21	9.33	55.77	11.19	55.50	11.39	0.278	0.808	0.485
LDH (U/L)	197.00	37.18	194.39	38.40	185.53	43.94	0.384	0.054	0.057
CRP (mg/dL)	0.14	0.19	0.15	0.26	0.28	0.72	0.579	0.289	0.310
LDL cholesterol (mg/dL)	109.75	21.20	119.59	26.96	118.51	24.47	0.019	0.669	0.061
A/G ratio	1.27	0.16	1.44	0.25	1.31	0.18	0.001	0.013	0.083
B/C ratio	29.22	9.70	32.61	8.76	28.71	8.43	0.013	0.013	0.870

A test for the purpose of obtaining diagnostic indicators of organic changes in living organisms by probiotics. All analyzes were performed using a paired *t* test, and mean and SD values are shown in each column. Abbreviations: BUN, blood urea nitrogen; TP, total protein; ALB, albumin; ALP, Alkaline phosphatase; AST, aspartate aminotransferase; ALT, Alanine Aminotransferase; T-BILC, total-bilirubin; GGT, Gamma-glutamyl transferase; HDL cholesterol, high-density lipoprotein cholesterol; LDH, Lactate dehydrogenase; CRP, C-reactive protein; LDL cholesterol, low-density lipoprotein cholesterol; A/G ratio, albumin/Globulin ratio; B/C ratio, BUN/Creatinine ratio.

## Data Availability

The data presented in this study are available in the article and Appendix A. The raw data are available on reasonable request from the corresponding author.

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
