# Peer review of "Different Reactions in Each Enterotype Depending on the Intake of Probiotic Yogurt Powder"

_microorganisms, 2021, doi:10.3390/microorganisms9061277_

Round 1

Reviewer 1 Report

The authors present an interesting study. Still some concerns need to be addressed:

Probiotics help women prevent bladder infections during menopause by increasing the presence of lactobacilli, a healthy microbiome that helps resist bladder infections in the absence of estrogen production. But please elaborate about in some paragraphs in the introduction.

In the introduction there is nothing mentioned about post-menopausa, its physiological characteristics and the need of the probotics, and terefore the aim of this study. Why the probiotics are recommanded in this stage?

Please mention about the differences between the 3 diets you reffer to in this study: Western, Korean and Mediterranean diet, and their benefits. What about their relation with intestinal microbiota profile, if any?!

What means mixed Korean diet?

Later on the manuscript stated: No participants consumed a Western diet. - why than you mention it?

"The abundance of Streptococcus, Agathobacter, and Christensenella, which are involved in lipid metabolism,showed a strong correlation with that of type P Prevotella, and triglyceride metabolism improvement was observed in all type P subjects. " - any differences reffering to the 2 diets consumed by the participants in this study?

In Table 1. The characteristics of participants in the baseline are presented, but nothing about the pros and cons about them. No theoretical and practical corelation between and how could help in the overall situation of the pacients in this study. For example: regular exercise is an excellent way to promote weight loss and overall physical health in this period, besides strength training exercises are recommended at least twice a week - in the table it is stated daily, and later on ightphysicalexercise such as walking for about 30 min every day, which may have a very good benefit! Please reffer to as a positive factor, comparing with literature (as no pacient in this case) with pacients no physical engagement.

In the conclusion: "In summary, we demonstrated that enterotyping of the gut microbiota can be useful for classifying individuals." and "Therefore, the intake of probiotics does not confer the same effects in all subjects, and it is necessary to considerthe enterotype." - please specify the category of the individuals as they are a special group!

Author Response

첨부 파일을 참조하십시오.

Reviewer 2 Report

This work is a pilot study aiming to assess the effects of a probiotic yogurt powder on intestinal microbiota and few clinical parameters. The authors have recruited 18 women, who had a 9 weeks follow-up including a 3-week period of probiotic yogurt intake powder (PYP).

First the authors found that PYP intake significantly improved stool shape. Then, by a Wilcoxon single rank sum test heat tree, they allowed the clear visualization of the modifications of microbial communities induced by PYP intake, and also of their reversion after the washout. The authors identified 3 enterotypes. In the B enterotype, they observed an improvement of blood glucose and ALP after PYP intake. In the P enterotype, they observed an improvement of TG levels. They showed that the PYP intake induced changes in intestinal microflora are specific of the enterotype. They fully characterized the microbiota of the 3 enterotypes and identified a precise pattern of about twenty correlated bacterial strains defining these enterotypes.

The manuscript is very well written. No detail is missing and all the results are described precisely and rigorously. The discussion presents numerous hypotheses coming from the results, which reflect the relevancy of the enterotype approach taken by the authors, and which are really worth exploring. The number of subjects is low but the approach by enterotype is very smart, and this give to my opinion an outstanding paper, which largely deserves its publication.

I have only few minor comments aiming to improve the manuscript:

In the figure 2, improvement of defecation activity, and abundance of beneficial bacteria were increased but the significance level was not reached. Even if this paper provides already a lot of data, would it possible to analyse these data, as well as stool type, by enterotype, to determine what is the influence of the 3 enterotypes on these parameters and if significant improvement can be highlighted in one of the 3 enterotypes.

In the figure 2, please make the legend of figure A right appear.

Line 325-326: “Enterotyping showed that only type P did not significantly change but AST improved after PYP intake”. Sorry, this sentence is not clear.

In the figure 5, the improvement of TG in R enterotype does not appear as reaching significance level, whereas in the result and in the discussion, this improvement is claimed. This discrepancy should be corrected.

Author Response

첨부 파일을 참조하십시오.

Round 2

Reviewer 1 Report

The authors improved the manuscript and adressed all suggestions/comments/remarks. The manuscript merits to be published. Literature is up to date.